# Degree of Polarization of High-Power Laser Diodes: Modeling and Statistical Experimental Investigation †

Alberto Maina [ID], Claudio Coriasso *[ID], Simone Codato and Roberto Paoletti

Convergent Photonics, Via Schiaparelli 12, 10148 Torino, Italy; alberto.maina@convergent-photonics.com (A.M.); simone.codato@convergent-photonics.com (S.C.); roberto.paoletti@convergent-photonics.com (R.P.)
* Correspondence: claudio.coriasso@convergent-photonics.com
† This paper is an extension of work originally presented in the European Semiconductor Laser Workshop, 17–18 September 2021.

**Abstract:** A statistical experimental investigation of the characteristic changes associated with the degree-of-polarization reduction of high-power laser diodes is reported. A simple model accounting for the stress-induced degree-of-polarization changes through the photoelastic effect is introduced to qualitatively support the experimental results. Functional characteristics addressed in the investigation are the threshold current, the slope efficiency, the polarization-resolved far field and near field, and the beam parameter product. Model outcomes and measured parameters related to different degree-of-polarization values have proven very useful for device optimization aimed to polarization multiplexing applications.

**Keywords:** laser diodes; semiconductor lasers; degree of polarization; polarization multiplexing; photo-elastic effect

## 1. Introduction

The polarization multiplexing (PM) of high-power laser beams is of utmost importance because it allows both their optical power and brightness addition [1,2]. Highly polarized input beams are, however, required by PM in order to avoid significant optical losses. Figure 1a shows a typical PM scheme in which the two input beams are combined by a polarization beam combiner (PBC), transmitting TE-polarized light and reflecting TM-polarized light. Since the two input beams have the same polarization state, TE in this case, one of them needs to be rotated by 90° through a half-wave rotator (HWR). Possessing a polarization state different from that required by the PM scheme, TM in this case, the input beam components are not routed to the output by the PBC and produce optical losses. Figure 1b shows a multi-emitter module fabricated with the high-power laser diode (HPLD) described in this work which exploits the PM scheme depicted in Figure 1a. This device is based on the PM of two optical beams produced by two sets of ten HPLDs each, spatially multiplexed using micro-optic elements. This multi-emitter device consistently emits output power up to 350 W at 976 nm on a 200 μm core and a 0.16 NA output fiber [2].

The Degree of Polarization (DoP) of the input optical beams represents the main figure of merit for the PM scheme depicted in Figure 1a. For mainly TE-polarized laser beams, like those emitted by the HPLD addressed here, this is defined as:

$$DoP = \frac{P_{TE}}{P_{TE} + P_{TM}} \tag{1}$$

where $P_{TE}$ and $P_{TM}$ are the TE-polarized and TM-polarized optical power, respectively.

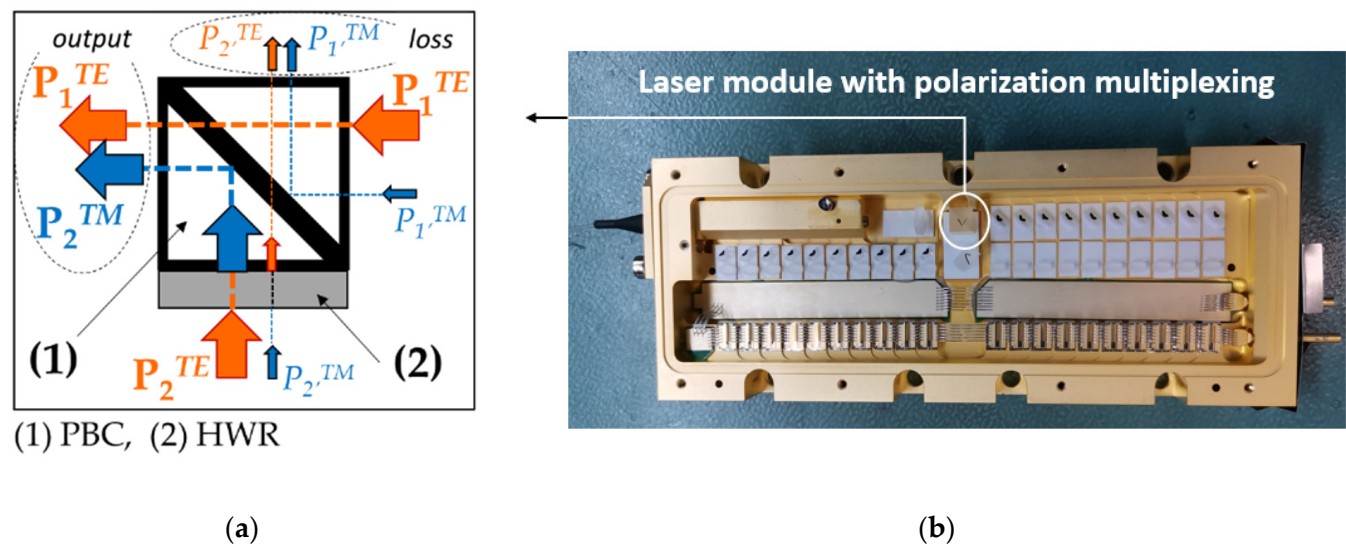

**(a)**  **(b)**

**Figure 1.** (**a**) PM scheme of two laser beams $P_1$ and $P_2$ with a PBC and an HWR. Thick shaded lines and arrows represent the main beam components while thin shaded lines and arrows represent the minor components. TE polarization is shown in orange while TM polarization is shown in blue. (**b**) 350 W multi-emitter module is based on PM of two laser beams obtained by two sets of the HPLD investigated here.

The intrinsic DoP of a HPLD can be very high, due to proper active material design aimed to the TE/TM gain ratio maximization. However, the DoP of mounted HPLD can be significantly reduced by unavoidable mechanical stresses [3,4]. Beyond the straightforward detrimental effect on the multi-emitter total power, the DoP being a main factor of the overall yield through the associate insertion loss, the DoP reduction implies less evident but important effects on the overall laser functional characteristics, which are the focus of this paper. Both the L-I characteristic (threshold current and slope efficiency) and the beam properties (near field and far field) are indeed significantly affected by DoP reduction, and the analysis of the relevant changes is of general interest for the investigation of stress-induced photo-elastic effect on laser diodes.

## 2. Material

The HPLDs addressed in this paper are broad-area semiconductor lasers with epitaxial structure similar to that reported in [5]. The epitaxial structure consists of a single 8 nm, 1.1% compressively strained $In_xGa_{1-x}As$ quantum well (QW) embedded between asymmetric, graded-index, separate-confinement heterostructure $Al_xGa_{1-x}As$ layers.

The QW and the epitaxial layer thicknesses, doping, and compositions were designed and optimized through the software tool Harold® by Photon Design [6]. High slope efficiency (>1.1 W/A), low series resistance (<20 mΩ), and high wall plug efficiency (>60%) were achieved with 4.5 mm-long Fabry–Perot laser diodes 95% HR/2% facet-coated with AR. Due to proper facet treatment, an output optical power in excess of 20 W was consistently achieved at 976 nm.

Figure 2a shows the vertical refractive index profile of the epitaxial heterostructure and the vertical section of the optical mode. Figure 2b shows the calculated optical gain as a function of photon wavelength $\lambda$ and at different carrier concentrations $N$ for TE and TM polarization, i.e., $G_{TE}(\lambda, N)$ and $G_{TM}(\lambda, N)$, respectively. The heavy hole ($e_1 - hh_1$) to light hole ($e_1 - lh_1$) energy gap displacement, enhanced by the QW compressive strain, produces a negligible TM gain compared to TE gain at laser emission wavelength, resulting in an intrinsic DoP of nearly 1.

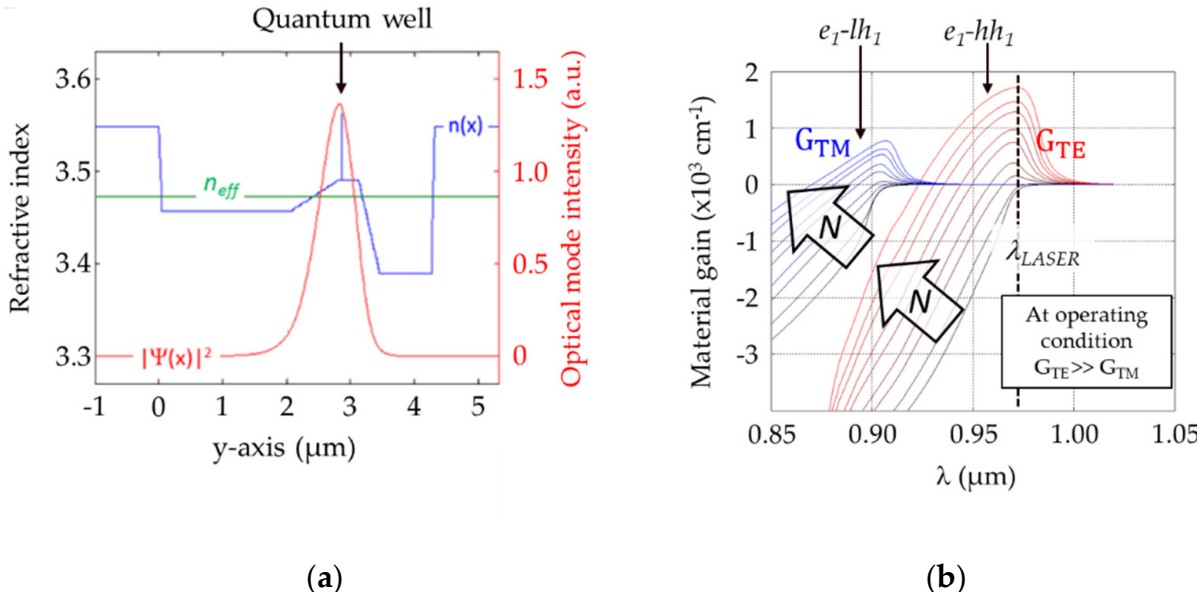

**Figure 2.** (**a**) Refractive index profile n(x), effective index value $n_{eff}$ and optical mode $|Y(x)|^2$. (**b**) TE and TM spectral optical gain calculated at different carrier density N ranging from $1 \times 10^{17}$ cm$^{-3}$ to $3 \times 10^{18}$ cm$^{-3}$.

The optical confinement of the broad area HPLD investigated in this work is defined by the epitaxial structure along the vertical direction, or fast axis (FA), and by a ridge waveguide along the horizontal direction, or slow axis (SA). A wide ridge width of about 200 μm is used in order to produce a power density at the output facet sufficiently low to avoid catastrophic optical mirror damage during device operation. The optical field is single-mode along FA and multi-mode along SA, with associated optical filamentation.

## 3. Model

The nearly 1 intrinsic DoP determined by compressively strained In$_x$Ga$_{1-x}$As QW material can be reduced by external mechanical stresses induced by chip fabrication (dielectrics, metals) and chip mounting on the carrier, through the photo-elastic effect [7–10]. Mechanical stresses are enhanced by chip morphology and can be particularly high on waveguide edges (see Figure 3 in which the distribution of *xy* stress component is calculated using a FEM simulation [10] performed with a COMSOL Multiphysics® tool). External stress effects on the polarization state of laser diode beams are well known; in the case of bulk diode lasers, in which both $e - hh$ and $e - lh$ optical transitions have the same energy gap, these effects were exploited to demonstrate polarization switching at different injection currents [11]. It is also worth noting that, according to our simulations and the literature [12–14] on compressively strained QW, external mechanical stress induces additional strain, affecting QW optical properties. However, the induced strain contribution to the QW, for the typical external stress induced by device fabrication and mounting, is significantly lower than the 1.1% epitaxial strain [12] and produces negligible effects on the DoP. The mechanical stress translates into an anisotropic refractive index change [15] through the photo-elastic tensor, which in turn determines the polarization change in the optical field within the active cavity, causing a DoP reduction [9]. The anisotropic refractive index of the semiconductor structure is due to both the quantum well active material and the optical waveguide. Starting from the material composition of each epitaxial layer and solving the Helmholtz equation, we can quantify the refractive index split between the TE mode, with electric field oscillating in the QW plane, $n_{xx}^0$, and the TM mode, with the electric field oscillating along the epitaxy growth direction, $n_{yy}^0$.

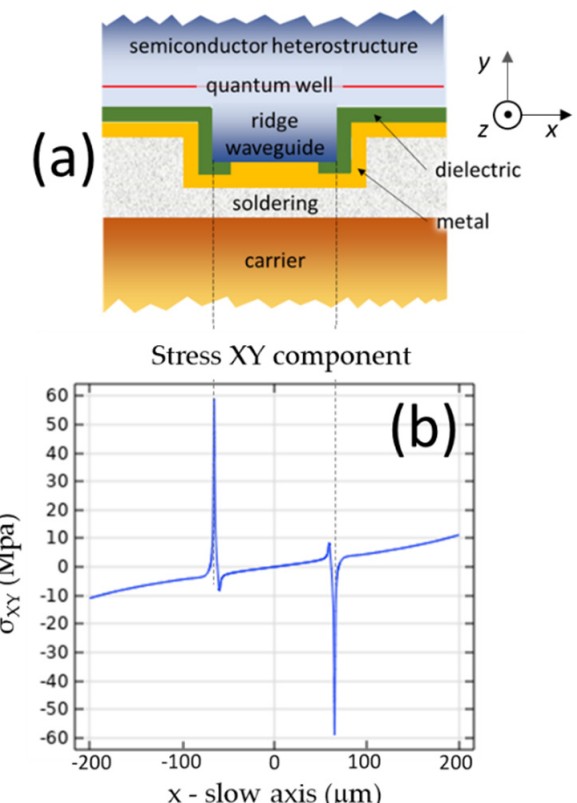

**Figure 3.** Stress calculation for a HPLD. (**a**) Schematic of upside-down mounted HPLD in which index guiding is achieved through a ridge waveguide. (**b**) 2D distribution of the stress tensor component $\sigma_{xy}$ obtained through a FEM calculation with COMSOL Multiphysics®.

It is useful to define the intrinsic anisotropy $\Delta n_{ii}^0$ due to the material structure [16]:

$$\Delta n_{ii}^0 = n_{xx}^0 - n_{yy}^0 \tag{2}$$

The laser cavity along the $z$-axis is defined through semi-reflecting mirrors at cavity facets. The TE-polarized light propagating along this direction is not affected by the intrinsic anisotropy $\Delta n_{ii}^0$. Furthermore, the small refractive index changes associated with the photo-elastic effect do not produce any significant change to facet reflectivity.

The anisotropic mechanical stress, due to the technological process and the mounting process, changes the refractive index, which can be conveniently represented as a tensor, through the photo-elastic effect. Refractive index components are related to the stress components through the following equation [15]:

$$\begin{pmatrix} n_{xx} \\ n_{yy} \\ n_{zz} \\ n_{xy} \\ n_{xz} \\ n_{zz} \end{pmatrix} = \begin{pmatrix} n_{xx}^0 \\ n_{yy}^0 \\ n_{zz}^0 \\ n_{xy}^0 \\ n_{xz}^0 \\ n_{zz}^0 \end{pmatrix} - \begin{pmatrix} C_1 & C_2 & C_2 & 0 & 0 & 0 \\ C_2 & C_1 & C_2 & 0 & 0 & 0 \\ C_2 & C_2 & C_1 & 0 & 0 & 0 \\ 0 & 0 & 0 & C_3 & 0 & 0 \\ 0 & 0 & 0 & 0 & C_3 & 0 \\ 0 & 0 & 0 & 0 & 0 & C_3 \end{pmatrix} \times \begin{pmatrix} \sigma_{xx} \\ \sigma_{yy} \\ \sigma_{zz} \\ \sigma_{xy} \\ \sigma_{xz} \\ \sigma_{zz} \end{pmatrix} = \begin{pmatrix} n_{xx}^0 + \Delta n_{xx}^s \\ n_{yy}^0 + \Delta n_{yy}^s \\ n_{zz}^0 + \Delta n_{zz}^s \\ n_{xy}^0 + \Delta n_{xy}^s \\ n_{xz}^0 + \Delta n_{xz}^s \\ n_{zz}^0 + \Delta n_{zz}^s \end{pmatrix} \tag{3}$$

where $n_{ij}^0$ are the unperturbed components, $\Delta n_{ij}^s$ are the stress-induced changes, and

$$C_1 = \frac{n_0^3(p_{11} - 2\nu p_{12})}{2Y}, \quad C_2 = \frac{n_0^3[p_{12} - \nu(p_{11} + p_{12})]}{2Y}, \quad C_3 = \frac{n_0^3 p_{44}}{2G} \tag{4}$$

The parameters $p_{ij}$ are the strain-optic constants; $Y$, $G$, and, $\nu$ are, respectively, the Young's modulus, the shear modulus, and the Poisson's ratio of the material layers.

From a mechanical point of view, we approximated the epitaxial structure with a single equivalent GaAs layer whose opto-mechanical properties are taken from literature and reported in Table 1. These properties are very different from all other materials considered in the simulations (dielectrics, metals) and similar for all the epitaxial layers, thus justifying the single equivalent layer approximation [17,18] aimed to strong computational time reduction.

**Table 1.** Opto-mechanical properties of GaAs.

| Constant | Value | Unit | Ref |
|----------|-------|------|-----|
| $p_{11}$ | −0.165 | - | |
| $p_{12}$ | −0.140 | - | [19] |
| $p_{44}$ | −0.072 | - | |
| $Y$ | 85.5 | GPa | |
| $G$ | 32.9 | GPa | |
| $\nu$ | 0.31 | - | [20,21] |
| $n_0$ | 3.47 | - | |

Considering a simplified one-dimensional approximation with light propagating along the longitudinal axis $z$, the Jones matrix formalism can be applied [22,23]. The light propagation for a polarized field can be described as:

$$\left( \begin{array}{c} \psi_x(z+dz) \\ \psi_y(z+dz) \end{array} \right) = e^{ik_0 ndz} \left( \begin{array}{c} \psi_x(z) \\ \psi_y(z) \end{array} \right) \tag{5}$$

where $\psi_x(z)$ is the TE field component, $\psi_y(z)$ is the TM field component, $k_0$ is the wavenumber, and

$$n = \left( \begin{array}{cc} n_{xx} - i\frac{\lambda}{4\pi}g_{xx} & n_{xy} \\ n_{yx} & n_{yy} - i\frac{\lambda}{4\pi}g_{yy} \end{array} \right) \tag{6}$$

is $\lambda$ the photon wavelength, $g_{xx}$ is the TE optical gain, and $g_{yy}$ is the TM optical gain, vanishing in our case ($g_{yy} \cong 0$).

The DoP can be calculated as:

$$DoP = \lim_{z \to \infty} \frac{|\psi_x(z)|^2}{|\psi_x(z)|^2 + |\psi_y(z)|^2} \tag{7}$$

by propagating the electric field using (5).

As shown in Figure 4, the DoP of the propagating field converges to a value defined by the stress-modified refractive index matrix, independently of the input polarization state, after a propagating distance

$$z \gg v_g \tau = max \, (g_{xx})^{-1} \tag{8}$$

where $v_g$ is the group velocity and $\tau$ is the photon lifetime within the laser cavity.

A useful and, to the best of our knowledge, original representation of the DoP as a function of the material properties (refractive index anisotropy $n_{xx} - n_{yy}$ and non-diagonal component $n_{xy}$) is reported in Figure 5. It is clearly visible that the DoP is maximized by increasing the refractive index anisotropy $n_{xx} - n_{yy}$ [16,24] and minimising the non-diagonal component $n_{xy}$. A white contour line at DoP = 0.95 is reported in the figure highlighting the critical region above it (low anisotropy, high non-diagonal component). Refractive index anisotropy $n_{xx} - n_{yy}$ can be enhanced by proper structure design, while $n_{xy}$ can be minimized by optimizing chip morphology, technological process, and chip mounting.

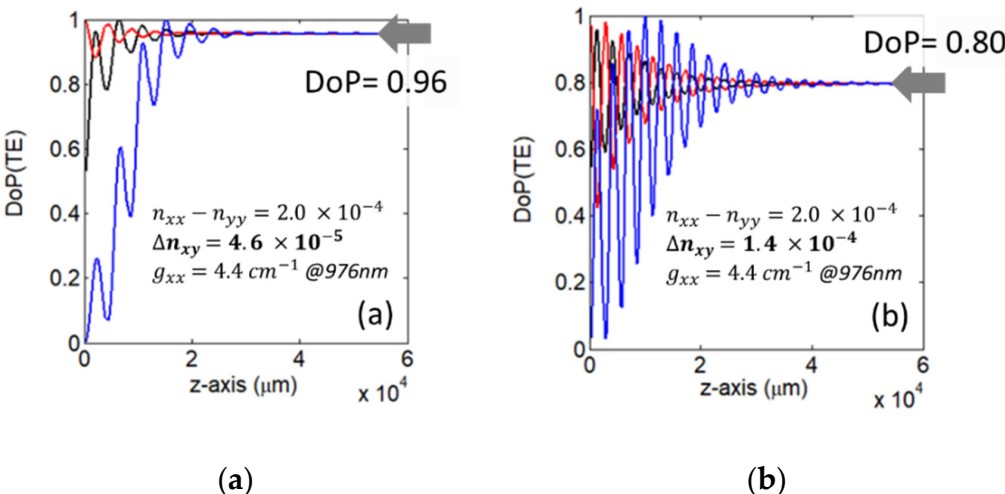

**(a)**                                                                **(b)**

**Figure 4.** DoP calculation starting with different polarization state of the input field $(\psi_x(0); \psi_y(0))$. Red curves refer to $(1; 0)$ (TE-polarized) input field, blue curves refer to $(0; 1)$ (TM-polarized) input field, and black curves refer to $(0.5; 0.5)$ input field. (**a**) Calculation performed with non-diagonal component of $4.6 \times 10^{-5}$, corresponding to small stress; (**b**) calculation performed with non-diagonal component of $1.4 \times 10^{-4}$, corresponding to high stress.

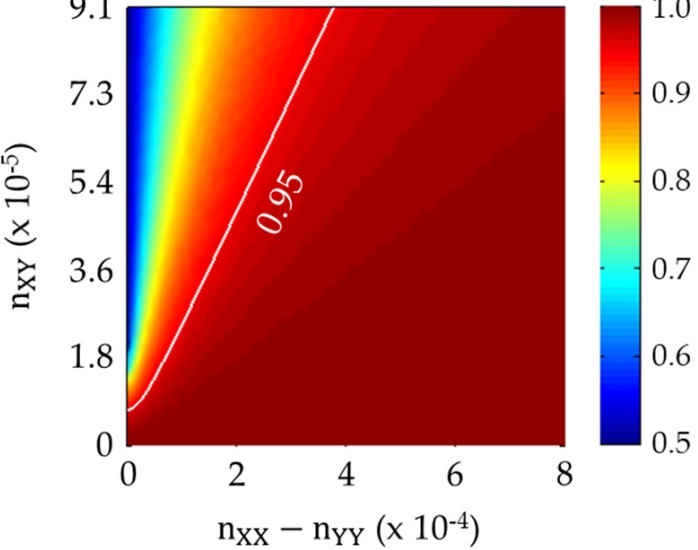

**Figure 5.** DoP calculation as a function of refractive index anisotropy, $n_{xx} - n_{yy}$ and non-diagonal component $n_{xy}$.

An example of the model capability for the analysis and optimization of the DoP of HPLD is reported in Figure 6, in which the effects induced by Cu layers embedded within the HPLD carrier for thermal dissipation improvement (see Figure 7) are addressed. Figure 6a shows that the stress has an effect on the non-diagonal refractive index component change $\Delta n_{xy}$, but also on the refractive index anisotropy change $\Delta n_{xx} - \Delta n_{yy}$, which significantly affects the refractive index anisotropy $n_{xx} - n_{yy}$, causing a DoP reduction (see Figure 5). A tradeoff between the intrinsic refractive index anisotropy of the HPLD, defined by the structure design, and the Cu layer thickness is necessary for DoP optimization, as clearly visible in Figure 6b, in which the high DOP region is above the white contour line corresponding to DoP = 0.95.

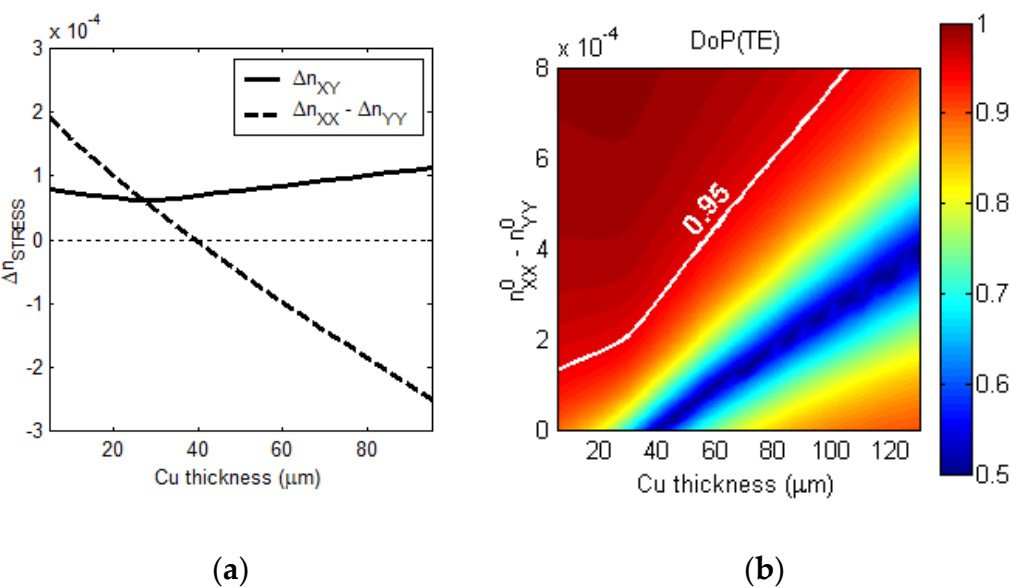

**Figure 6.** Effect of the thickness of Cu layers in the carrier: (**a**) refractive index component changes (**b**) DoP as a function of the Cu layer thickness and the intrinsic refractive index anisotropy.

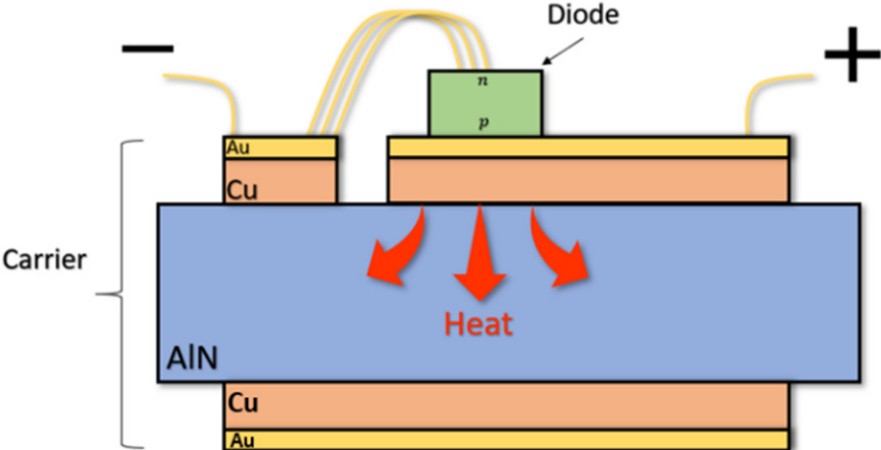

**Figure 7.** Schematic of a HPLD mounted on a carrier embedding Cu layers.

The model introduced in this section was used to predict and analyze the DoP reduction of HPLD due to different technological steps, including chip fabrication and chip mounting on carriers. It is particularly useful for chip layout and carrier structure optimization. In the following section, the experimental investigation of DoP and the changes due to its reduction on the functional characteristics of HPLD (threshold current, slope efficiency, near field, far field) will be described.

## 4. Experimental Results and Discussion

As discussed in the previous section, mechanical stress causing the DoP reduction is enhanced by chip morphology and can be particularly high on waveguide edges; the result of COMSOL Multiphysics® simulations for the stress is shown in Figure 3b. The resulting polarization change in the optical field is equivalent to an internal loss producing an increase in the laser threshold current [25] and a decrease in the slope efficiency. See Figure 8, in which measurements of two sets of HPLD mounted with different die-bonding conditions are shown.

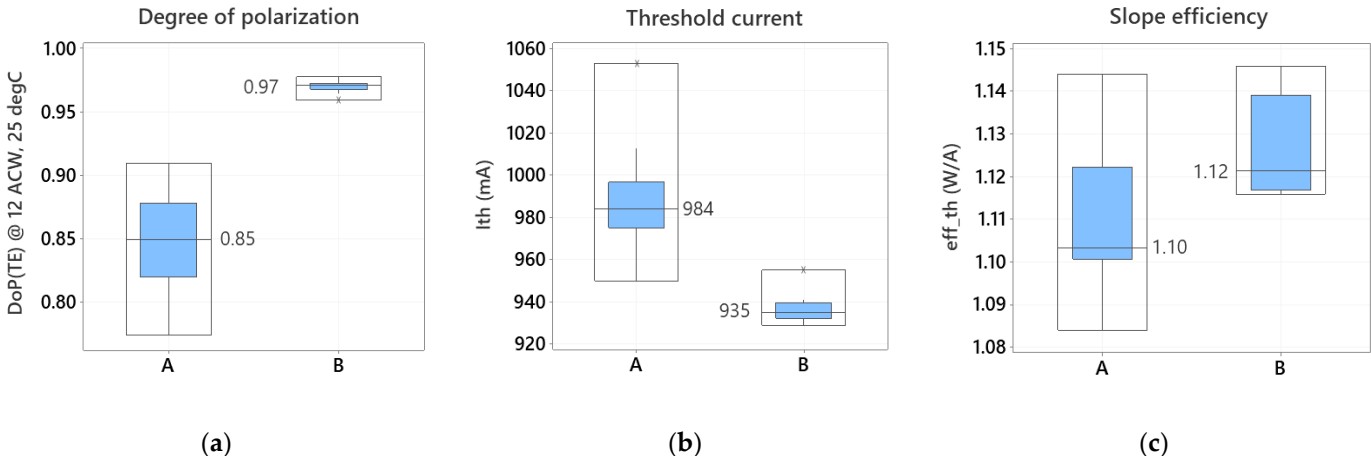

**Figure 8.** Measured effects of stress-induced DoP reduction on functional characteristics of two sets of HPLD mounted on carriers. Set A represents 10 high-stressed HLPD while set B represents 10 low-stressed HLPD. (**a**) Box plots of measured DoP; (**b**) box plots showing threshold current increase from B to A; (**c**) box plots showing slope efficiency decrease from B to A.

Two-sample *t* tests showed significant differences between the two sets, with *p*-value lower than 0.05 for all of the functional characteristics reported in Figure 8.

The laser beam shape is also significantly affected by DoP reduction. Figure 9a shows the TE-polarized slow axis far field (FFSA) increase by about 9% and the TM-polarized FFSA decrease by about 2% for HPLD with different DoP from 0.97 to 0.85. The beam parameter product, which characterizes the beam quality and is defined as the product of half-beam divergence and half-beam width, is also significantly increased by DoP reduction. See Figure 9b.

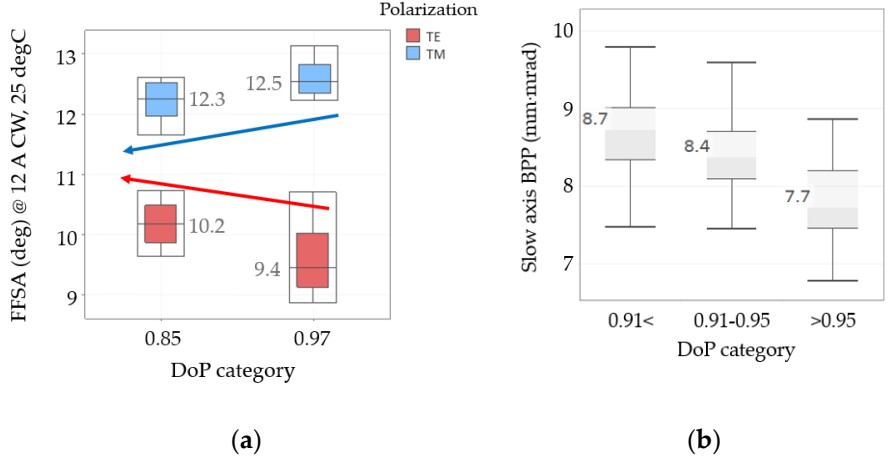

**Figure 9.** Measurements of HPLD with different DoP. (**a**) Polarization-resolved slow axis far-field changes for 8 HPLD with DoP = 0.85 and 11 HPLD with DoP = 0.97; (**b**) slow-axis BPP changes for HPLD divided into three DoP categories (93 with DoP < 0.91, 124 with $0.91 \leq \text{DoP} \leq 0.95$, and 64 with DoP > 0.95).

For all of the functional characteristics reported in Figure 9, two-sample *t*-tests showed significant differences with *p*-values lower than 0.05.

Experimental setup for the DoP measurements and the polarization-resolved near-field measurements reported in this section are shown in Figure 10. Far-field measurements were performed on a standard goniometric radiometer setup.

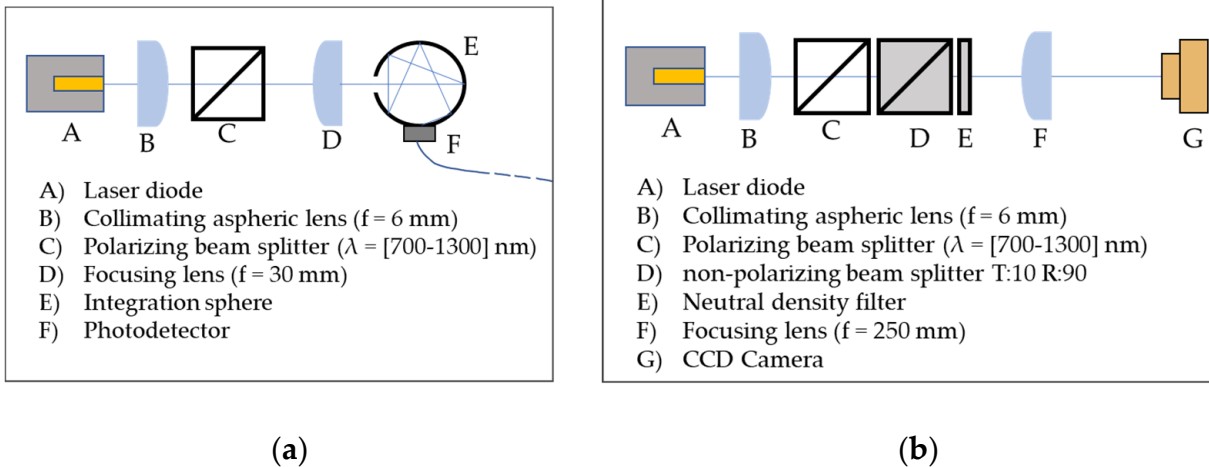

**(a)**           **(b)**

**Figure 10.** Measurement setup of (**a**) DoP; (**b**) slow axis near field.

Figure 11 shows the polarization-resolved near-field slow axis (NFSA) for 324 HPLD devices with different DoP, ranging from 0.86 to 0.98. In particular, the TM-polarized NFSA reported in Figure 11a shows sharp peaks at the waveguide edges [3,26] when the DoP is high and spreads to the whole waveguide, about 200 μm wide, when the DoP is decreasing. At the same time, the TE-polarized NFSA, reported in Figure 11b, shows an increase in optical filamentation with DOP decreasing. These effects, due to different mode excitation of the multimode lateral waveguide, are highlighted in the top color maps with the black shaded lines dividing regions of different behavior for high DoP (H) and low DoP (L), as a visual guide. The bottom images refer to single NFSA acquired for low-DoP and high-DoP devices, and clarify the NFSA shape discussed above.

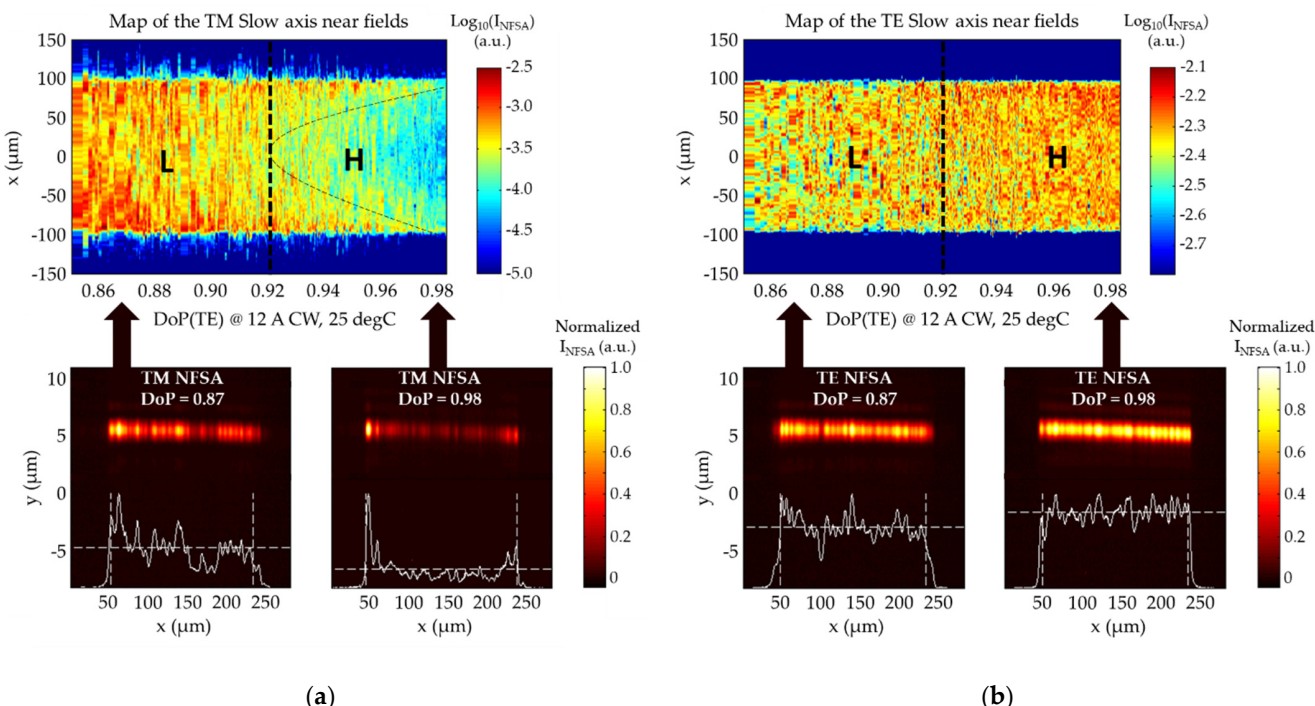

**(a)**           **(b)**

**Figure 11.** Measurement of 324 mounted HPLD: (**a**) TM-polarized NFSA; (**b**) TE-polarized NFSA.

For mainly TE-polarized compressively strained-QW HPLD, the TM-polarized NFSA represents a distribution of the mechanical stress on the device and can be very useful in the development of high-DoP HPDL, by optimizing chip dielectrics, metals, shape,

mounting process, and carrier structure. Given that mechanical stress is an effect of the randomly changing technological conditions within a statistical process control, a statistical approach to the measurements is of utmost importance. All of the images shown here refer to statistical measurements of ideally equivalent HPDL devices and the results' significance was demonstrated using a statistical inference test. Moreover, the NFSA map representation of many devices with DoP randomly distributed over a wide interval, such as those reported in Figure 11, allows a clear observation of different behaviors and the transitions among them, highlighted by the dashed lines, whose identification would be very difficult without a statistical population of devices.

## 5. Conclusions

DoP reduction effects on electro-optic and laser beam properties of HPLD were extensively investigated using a statistical approach on hundreds of devices. Measured results were validated by statistical inference tests and, by identifying and reducing residual mechanical stress, proved to be very useful in high-DoP device optimization. The analysis is supported by the model introduced in Section 3, which produces the DoP value starting from the tensor stress components calculated for the HPLD structure though a FEM modelling. To the best of our knowledge, the effect on the DoP of HPLD due to the refractive index anisotropy, together with non-diagonal stress, and in the presence of optical gain, was originally addressed in this work by introducing a convenient representation. Furthermore, the uncommon statistical approach in the experimental DoP investigation permitted an in-depth analysis of functional characteristic changes.

**Author Contributions:** Conceptualization and methodology, A.M., C.C., S.C. and R.P.; software and analysis, A.M.; investigation, A.M., C.C. and S.C.; writing, A.M., C.C. and S.C. All authors have read and agreed to the published version of the manuscript.

**Funding:** This research received no external funding.

**Institutional Review Board Statement:** Not applicable.

**Informed Consent Statement:** Not applicable.

**Data Availability Statement:** Not applicable.

**Acknowledgments:** The authors would like to thank the entire Diode Fab staff at Convergent Photonics for their technical assistance with the fabrication, assembly, and characterization of the laser diodes presented here.

**Conflicts of Interest:** The authors declare no conflict of interest.

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
