# Peer review of "Degree of Polarization of High-Power Laser Diodes: Modeling and Statistical Experimental Investigation†"

_applsci, doi:10.3390/app12073253_

Round 1

Reviewer 1 Report

The manuscript by Maina et al. makes an impressive contribution to the scientific understanding of the packaging of high power laser diodes from the perspective of the employees of a high technology company. It is shown that the degree of polarization of these devices is indeed determined by the packaging technology. Therefore, we strongly endorse the publication of this work.

The modeling corresponds to the current state of the art and does not contain any novelties from our point of view. It is relatively broad and could be shortened. The experimental material is deeply impressive since data from hundreds of devices are summarized here in a systematic way. Regrettably, the presentation of the experimental results is totally inadequate.

There are no descriptions of the measurement apparatus or methodology at all. This does not necessarily have to be detailed or could be covered. By citations as well.

There are a number of technical deficiencies. This starts with the numbering of the figures, figure 5 and 6 exist twice each. Many abbreviations are introduced that are never used at the end. The figure captions are not sufficient and do not allow the full understanding of the results shown. In Fig. 7, this is quite extreme. Captions on the images (e.g. axis) are partly so small that they are illegible.

We suggest a complete revision of the paper in which the experimental results must . Be explained in detail. We consider the results shown in Figure 7 as central for the paper and expect an appropriate presentation. 

Author Response

see attached document

Reviewer 2 Report

Review of the manuscript "Degree of polarization of high-power laser diodes: modeling and statistical experimental investigation"

The authors' study aims to identify the layer and process design for a diode laser with the highest possible degree of polarization. The technical importance of a high degree of polarization will be worked out using the example of polarization multiplexing, but should also be adequately known in the laser community. An epitaxial structure optimized for 976 nm is chosen as the basis for the investigations. Starting from this structure, the material values are used to infer the change in the DoP by considering mechanical stresses and the changes in the direction-dependent refractive index indicated by these stresses. Characteristic fields are determined - once as a function of the stress and once as a function of the Cu layer thickness in the heat sink - which make the tendencies of the DoP change visible over a wide parameter field.

To verify the results of the simulation, experiments are performed on the lasers in the second part of the manuscript. Unfortunately, this part seems somewhat unfinished and lacks concrete references to the simulations. In detail, the following is noticeable here:

  • the experimental setup (e.g., measuring equipment) is not described
  • it is not stated what is "A" and "B" in figure 5, thus the statements about the changes between the two test groups are without meaning
  • figures 5 and 6 appear twice in the manuscript, here also the references from the text are mixed up
  • if the title of the manuscript refers to a "statistical" study, at least the number of measured lasers should be given.
  • for a convincing verification of the simulation results, a content-related connection must be established, i.e., which predictions (qualitative and/or quantitative) from the simulation could be proven experimentally; possibly this connection is behind the sample groups "A" and "B", but this is not explained
  • especially in the context of figure 7 many data are given and abbreviations are introduced, but the explanation of the content (especially in section 5. Conclusions) has to be developed further

Overall, towards the end, the manuscript seems incomplete and somewhat like a presentation of data without a complete concept of content. This should be revised.

Additional Notes:

  1. The concept of changing optical properties of semiconductors and especially of diode lasers under mechanical strain is well known and described in the literature (e.g. in the book "Quantum-Well Laser Array Packaging", McGraw-Hill Education (publisher), 978-0-07-166164-5 (ISBN), J. Jimenez and J.W. Tomm) and the optical fundamentals of the effect on the refractive index have also been described analogously to the manuscript (e.g. in the publications of Daniel T. Cassidy). Here it would be helpful if the authors could integrate the knowledge gained by their work again in the style of a statement of novelty in the introduction.
  2. Please check that there is always a space between numbers and units (e.g. page 2, line 59 and page 3, line 73).
  3. Regarding the model for the occurrence of TM polarization: Is there also a significant influence due to the lattice strain already at the generation of the radiation in the QW? Here, optical transitions are possible by the tensions, which generate TM-polarized light, which are actually forbidden by the quantum-mechanical selection rules in the idealized QW.
  4. Page 4, Lines 110ff : If a single GaAs layer is assumed here instead of epitaxy, are the values used mean/weighted values? Are the values adjusted again to the current structure or just taken from the literature sources?
  5. Page 5, Line 132: “[…] propagating the electrical field using (6)” Is the reference to equation (6) correct? Since this only specifies how the refractive index is determined from the dielectric constant.
  6. Caption of Fig. 4: Shouldn't equation (10) rather than (7) be referenced here?
  7. Caption of Fig. 4: That there is no influence of the input polarization on the result, however, is only valid under the condition of equation (11). This should be noted here, because there are clear differences in the left part of the graph and the statement is only valid in the limiting case.

Author Response

see attached document

Reviewer 3 Report

This is an interesting article but I believe than can be improved. A brief guide is the following:

  1. Page 4, line 99 it is mentioned that: "The laser cavity along the z-axis is defined through semi-reflecting mirrors at cavity facets." But it is not discussed whether, or in what way, the reflectivity is affected by the stress.
  2. Page 4, line 110 it is mentioned that: "From the mechanical point of view, we made the approximation that the epitaxial structure can be described by a single equivalent GaAs layer, whose opto-mechanical 
    properties are reported in table 1".  It is important whenever an approximation is made, to discuss how good the approximation is as well as its limitations.
  3. Page 5, line 134, says: "It is worth noting that the DoP result doesn’t depend on the polarization of the input field".  I would suggest you the following wording: "Due to (your explanation) it is worth noting that the DoP result doesn’t depend on the polarization of the input field"
  4. Page 5, line 141, says: "If the gain is neglected in the calculation the convergence can never be achieved".  It is not clear about what gain are you talking, I suggest a clearer explanation.
  5. Page 5, lines 146-148. I suggest you to clearly state whether this is a new result, otherwise provide additional references.
  6. Page 8, line 196, says: "It is worth underlying that, being the mechanical stress a random effect".  How do you precisely characterize this randomness?
  7. Fig 2 is a bit confusing.  Even though it is clear that it is a qualitative diagram, the thickness of the arrows is changed along its path and the reader may assume that this is due to some sort of loss. (This is not a very important observation but I believe that you could improved it)
  8. Fig 6.  I also believe that the discussion of this figure could be improved.  This I believe is important.

Author Response

see attached document

Round 2

Reviewer 1 Report

The manuscript has been improved.

Reviewer 2 Report

I would like to thank the authors for answering my questions and implementing the comments by making changes to the text. I think the manuscript is sound in its current form.